# Towards a Greener University: Perceptions of Landscape Services in Campus Open Space

**Carla Ana-Maria Tudorie** [1,*] **, María Vallés-Planells** [1] **, Eric Gielen** [2] **, Rosa Arroyo** [3] **and Francisco Galiana** [1]

1   Department of Rural and Agrifood Engineering, School of Agricultural Engineering and Environment, Universitat Politècnica de València (UPV), 46022 València, Spain; convalpl@agf.upv.es (M.V.-P.); fgaliana@agf.upv.es (F.G.)
2   Department of Urbanism, School of Civil Engineering, Universitat Politècnica de València, 46022 València, Spain; egielen@urb.upv.es
3   Department of Transport, School of Civil Engineering, Universitat Politècnica de València, 46022 València, Spain; maarlo3@cam.upv.es
*   Correspondence: cartu@upv.es

**Abstract:** Universities are showing a growing interest in becoming green institutions and improving campus open space management. Well-designed urban landscapes guided by green criteria integrate eco-friendly infrastructure which may be effective in facing urban challenges in the context of climate change. Student preferences and uses of campus outdoor environment should draw the attention of campus landscape planners. This study aims to analyse how the university community perceives landscape services provided by the Spanish Universitat Politècnica de València's campus open space. An online questionnaire was sent to the university community to check its opinions, level of satisfaction, and their demands related to the current situation of the outdoor areas. Campus open spaces with different urban green infrastructure have a high potential to provide cultural, provisioning, and regulation landscape services. Respondents perceive the main benefits provided by campus open spaces to be that they are a place to relax, meet friends, and pass through. Their needs related to the welfare of outdoor areas and their preferences differ according to age, occupation, and time spent at the campus. This paper intends to help the university to meet environmental guidelines and to help other universities in their endeavour to reach sustainability and ensure the university community's well-being.

**Keywords:** landscape services; university community; outdoor environment; urban green infrastructure; design guidelines; Universitat Politècnica de València; Green Flag Award

## 1. Introduction

This paper refers to research conducted at the district level and focusses on the potential of university campuses to be sustainable green spaces representative of the university community and even of a town or city. Sustainability criteria should guide how campus open spaces (COS) are designed. The green look and the quality of campus outdoor environments are the materialisation of a proper implementation of sustainable requirements in a higher education environment.

First, due to their large surface areas, they may be suitable both for providing a learning environment and creating a complex and dynamic semi-natural ecosystem. Secondly, a good COS design promotes the formal indoor learning process through casual encounters, leading to social interactions and the exchange of ideas [1]. Campus open spaces are understood as accessible urban ecosystems, without buildings [2] and made up of different elements connected to each other: green spaces and grey spaces [3].

University campus green spaces have recently earned a place in the ranking of important urban green areas, receiving more attention than in the past [4,5]. Universities' efforts to attain sustainability are usually reflected in the level of awareness, knowledge, and interest in sustainability issues, students' perception of sustainability and a green design for campus open spaces. At present, there are many universities that are endeavouring to design or improve their campus, creating the characteristics and functions of an urban ecosystem, to gain a "garden-park" look. Many universities would like to become the world's greenest universities, e.g., University College Cork, Trinity College Dublin, and the University of York. Aspiring candidates should have attractive university facilities [4] and apply eco-friendly principles for the well-being of the university community, city, and natural environment. In 2018, the campus of the Universidad de Navarra became the first Spanish university campus to receive the Green Flag Award [5]. Following the award's established criteria, the Universitat Politècnica de València (UPV) has developed outdoor environment guidelines to obtain well-managed open and green spaces [6], defined by sustainable design [7]. Since 1997, the UPV has had an Environmental Policy in place. It was the first Spanish university to implement an Environmental Management System [6].

In this study campus open spaces are considered multifunctional landscape services (LS) providers, which simultaneously offer vital benefits, such as improving air quality, ensuring pleasant climatic conditions, habitats for biodiversity, offering outdoor spaces to play, do sport, rest, and provide psychological benefits. Although "biodiversity" encompasses all kind of species (including small animals) and also habitats, the aims and the scale of this study makes us consider only the superior plant vegetation, due to their apparent and easily recognisable structure in the campus.

Landscape services are the output of natural and other forms of capital. Because COS provide some particular benefits which are a product of the current university community's activities, the concept of LS is more applicable to a campus' outdoor environment than to ecosystem services, which are exclusively derived from natural capital [8].

There is work done in the field of use patterns, preferences, and students' perceived quality of life in campus green spaces and campus open space design. The results of research from universities on four continents (Europe, North America, Asia, and Australia), developed in countries including the United Kingdom [9], the United States [10,11], Turkey [11], Israel [12], Vietnam [13], China [14], and Australia [14], have contributed to the literature on campus outdoor areas.

Cooper and Wischemann [1] propose a campus outdoor area typology according to a specific design and distribution around the university campus, such as the home base or spaces adjacent to specific buildings, e.g., the front porch, back door, front and back yards, and the common turf or outdoor spaces used by everyone, like the main plaza, campus entries, and favourite green areas. Each element is described and associated with different activities and uses. Abu-Ghazzeh [12] and Hanan [13] use the same typology in their research, confirming that campus outdoor spaces primarily provide a social, study, meeting, and eating environment, and secondarily a quiet, calm, relaxed, serene, green, and comfortable space, needed to maintain mental health and reduce stress. Gulwaldi's findings [11] associate the green campus' open spaces with green restorativeness and student quality of life. Abu-Ghazzeh [12] and Speake et al. [9] reveal users' preferences for different types of COS, like remote natural areas (students who avoid crowded places), areas with urban aspects (popular students), and the aesthetics of tamed and manicured landscapes (which denote a lack of ecological awareness). Concerning visual quality assessment (VQA) in the university campus environment, Polat et al. [15] demonstrate that visual quality increases in the areas where naturalness and plants are dominant, especially trees and shrubs, which also increases the students 'preference for landscapes.

Regarding the perceptions of campus open spaces, Abu-Ghazzeh [12] records the relationship between the physical features of students' favourite open spaces and the use of campus spaces. McFarland et al. [10] find a relationship between the frequency of using campus green spaces and perceived quality of life. Speake et al. [9] discover a connexion between students' perceptions of the quantity of campus green space and campus green space awareness.

However, there are few works that specifically refer to perceived landscape services in urban open space. Vallés-Planells et al. [16] explore the perceptions of socio-cultural landscape services which have been implemented in a set of urban open spaces in Ghent, Belgium, and Valencia, Spain. The campus of the Universitat Politècnica de València (UPV) is one of the study plots where respondents often identified the provisioning of opportunities for learning and for social encounters, and the contribution to place identity. Lau et al. [14] run an in-depth analysis of three design strategies to achieve healthy campus open spaces. Their research is quite possibly the only reference to the capacity of the campus to provide socio-cultural or landscape services. However, the study of Lau et al. [14] contains only design recommendations and not direct assessments of the perception of open space benefits. This knowledge would be useful to inform the design of campus green spaces in terms of the provision of cultural services.

The main aim of this paper is to discover the university community's perceptions of open spaces and landscape services provided by the campus outdoor environment, within the setting of UPV's campus. Its objectives are:

1.  To discover the level of satisfaction and needs of university respondents towards all campus open spaces (COS).
2.  To assess how the university community perceives the quality with which landscape services (LS) are provided by campus open spaces, considering the current state and management of the UPV's outdoor areas.
3.  To explore whether there are significant differences between users regarding preference by the types of COS and LS provided; to discover how university community members perceive COS with different urban green infrastructure (UGI), from the point of view of attributes (items) and the capacity of open spaces to provide LS.

This study will contribute to decision-making in campus space design by connecting the existing literature on campus outdoor environment with new assessments of users' perceptions that reveal the capacity of campus open spaces to provide landscape services.

## 2. Methodology

### 2.1. Scope of Study

The study area is the campus of the Universitat Politècnica de València, a Spanish public university located in Valencia (Figure 1). Valencia is the capital of the autonomous community of Valencia and the third largest city in Spain, after Madrid and Barcelona, with around 800,000 inhabitants. Valencia is located on the Spanish Mediterranean coast on an alluvial plain, formed through a repeated depositional sequence of the Turia River. This city has a Mediterranean climate with long, hot, and dry summers and mild and wet winters [17].

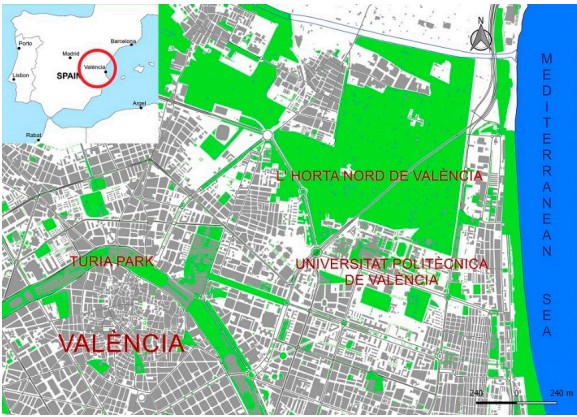

**Figure 1.** Location of the campus of Universitat Politècnica de València.

The campus of Universitat Politècnica de València covers almost 57 hectares. Approximately 22% of the total surface area of the campus is occupied by green space. More than eighty buildings, around 87% of the built-up area, are situated around Plaza Agora, considered the heart of the university [18]. The campus is situated in the neighbourhood of L'Horta de València and is the new northern urban limit of the city and a transitionary space between urban and rural environments (Figure 2). L'Horta de València forms a unique landscape integrated into a well-known Mediterranean orchard. This singular agricultural system was declared a protected landscape and part of the city's cultural heritage [19,20] and has been recently (2019) recognised as a Globally Important Agricultural Heritage System (GIAHS) and as a 1200-year-old productive farming system, in which agricultural and hydraulic heritage and culture are integrated [21].

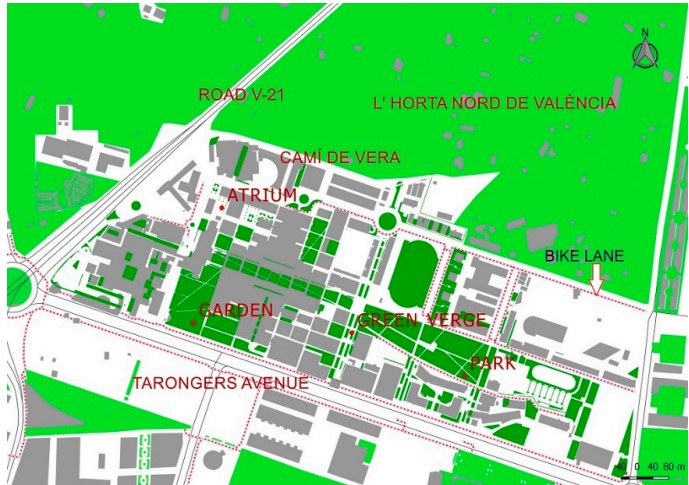

**Figure 2.** Distribution of campus open spaces (COS) in the Universitat Politècnica de València (UPV)'s campus.

The university community consists of 36,483 members [22] distributed over 13 schools, 42 departments, 34 research institutes, and 87 services. There are 28,801 students, 4971 employees (teachers and administrative staff), and 2711 external personnel. All were invited to voluntarily participate in an anonymous online survey conducted for this paper.

The campus open spaces include the Campus Botànic UPV, where approximately 2300 trees from 251 different species from five continents create the botanical heritage of the campus. The species have been classified into bushes (31.87%), deciduous (26.29%), evergreen (12.75%), and semi-evergreen trees (12.75%), herbaceous plants (10.36%), conifers (6.37%), palm trees (6.37%), and other species (5.98%) [23].

### 2.2. Landscape Services Classification

This study investigates landscape services (LS) provided by campus open spaces (COS). The framework developed by Vallés-Planells et al. [24] is taken as a reference to classify campus landscape services. The landscape services considered for this study are presented in Table 1.

**Table 1.** Classification of landscape services according to Vallés-Planells et al. [24].

| Theme | Class | Group | Landscape Services |
|---|---|---|---|
| Provisioning | Daily activities | Circulation setting | Outdoor place to pass through |
| | | Work/study setting | Outdoor space to work or study |
| | | Basic needs setting | Outdoor space to eat or rest |

**Table 1.** *Cont.*

| Theme | Class | Group | Landscape Services |
|---|---|---|---|
| Regulation and maintenance | Flow regulation | Water flow regulation | Space that contributes to flooding control and reduction |
| | Regulation of physical environment | Atmospheric regulation | Space that improves air quality |
| | | | Space that helps to provide pleasant climatic conditions (temperature, humidity, solar radiation, ventilation) |
| | Regulation of biotic environment | Lifecycle maintenance and habitat protection | Space that maintains and increases biodiversity |
| Cultural and social | Health/enjoyment | Active enjoyment (physical health) | Outdoor space to do sport or walk |
| | | Passive enjoyment (mental health) | Outdoor space to disconnect or relax |
| | Self-fulfilment | Didactic resources | Outdoor education |
| | | Scientific resources | Outdoor sites for research |
| | | Source of inspiration | Source of inspiration for art (photography, cinema, painting, sculpture, etc.) |
| | Social fulfilment | Place identity | Space that contributes to the establishment of psychological and cultural connections between the university community and the campus |
| | | Social interactions | Space that offers meeting places for social interaction |

## 2.3. Survey Tool

This survey used a questionnaire to collect information on the study variables. Quantitative data were gathered using an online survey distributed by e-mail and social media in Spanish to the university community. Open-ended, semi-open-ended, and closed-ended questions were used.

The questionnaire was structured in three sections: Section 1 is about preferences and the evaluation of campus open spaces; Section 2 deals with LS perceptions related to the current state and importance of campus open spaces; Section 3 aims to define the users' socio-demographic characteristics. Trap questions were used to identify those respondents who were not paying attention to the questions, which could bias the whole survey with sub-optimal responses. Pilot testing was conducted and validated to ensure the study's consistency.

At the beginning of the questionnaire, the definition of campus open spaces, the objectives of the study, and data privacy statements were added to clarify the goal of the study. There were no incentives for the participants. A reminder to participate in the survey was published on social media and sent by e-mail.

## 2.4. Questionnaire Structure

The first section focuses on user preferences for COS. It was developed based on previous research which focused on the frequency of visits to campus green and open spaces [25–28].

Colour photographs are commonly used in studies of perceptions and judgement of the visual environment [12,26,29,30]. For this study, four images of different UGI elements were included. Open spaces with different green cover and permeable areas were selected: atrium, garden, park, and green verge (Figure 3). These were selected to represent the different building stages of the

university campus. Atrium is the newest campus area, garden was included in the initial design of the campus, and the other two elements were built in the intermediary stage. Moreover, they portray the most popular areas in the campus and meet the condition of including different elements of urban green infrastructure. Four photographs are the minimum sample to cover the variability of the campus outdoor environment regarding the provided landscape services. For the classification of UGI, EU criteria [31] were used. Their organisation by location and use, inspired by Cooper and Wischemann's study about COS [1], is obtained from the UPV campus map, developed with GIS, combining an aerial photograph updated in 2018 (25 cm resolution) and the campus' cadastral map.

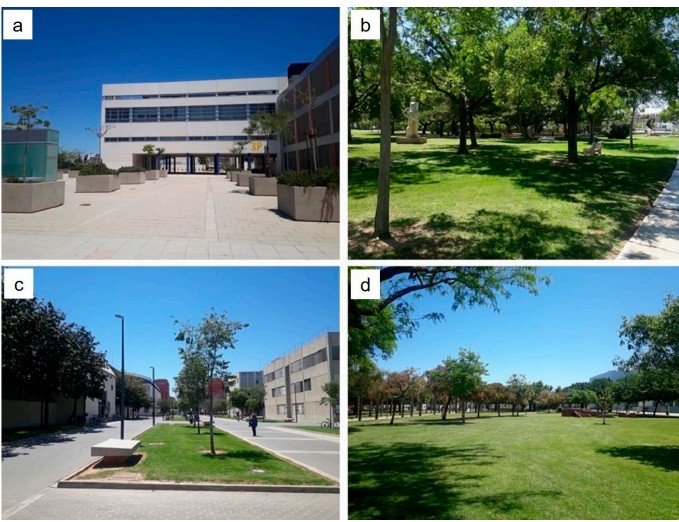

**Figure 3.** The four types of urban green infrastructure (UGI) elements selected: atrium (**a**), garden (**b**), green verge (**c**), and park (**d**).

The pictures used in the questionnaire were taken from different locations to cover the diversity of COS available at the UPV (Figure 3), including permeable and impermeable surfaces and different species and strata diversity (Table 2).

**Table 2.** Characterisation of COS types.

| Ecological Parameters | Atrium | Garden | Green Verge | Park |
|---|---|---|---|---|
| Permeable cover (%) | 8.0 | 98.0 | 40.0 | 96.0 |
| Impermeable cover (%) | 92.0 | 2.0 | 60.0 | 4.0 |
| Species richness (number of vegetation species) | 6 | 21 | 2 | 16 |
| Strata diversity (number of strata) (0–4) | 2 | 4 | 1 | 3 |
| Shannon–Wiener Index (trees) (0–4) | 0.6 | 2.5 | 0.7 | 2.3 |
| Maximum Diversity (trees) (0–4) | 0.7 | 2.8 | 0.7 | 2.7 |
| Classification of COS (Cooper and Wischemann) [1] | outdoor | outdoor | Common | common |

According to Cooper and Wischemann's classification [1], "atrium" and "garden" are outdoor areas adjacent to specific buildings, while "park" and "green verge" are in the campus' common areas. All open spaces are pass-through areas, but garden and park are multifunctional outdoor areas, where the university community engages in more types of activities, like meeting, eating, studying, resting, etc., due to their isolation and a higher degree of permeability and greenness than the other two COS (Table 2).

Based on these images, participants were asked:

1. To rate the COS on a five-point Likert scale (from 1, "I do not like it at all", to 5, "I like it a lot") [32].
2. To assess six statements [30] related to the following attributes: "It is beautiful"; "It is natural"; "It is maintained"; "It is useful for the management of environmental functions (biodiversity,

permeability, climate comfort, $CO_2$ capture)"; "It is important for the image of the university campus. The answers were rated on a five-point Likert scale (1-strongly disagree, 2-do not agree, 3-neutral, 4-agree, and 5-completely agree) [32]; and

3.  To choose landscape services provided by the campus outdoor areas used by everyone, such as park and green verge.

Section 2 aims to establish the perception of the campus open spaces. Measuring perceptions cannot be done with only one method. So, a more complex instrument for measuring perceptions of all three LS themes (provisioning, regulation and maintenance, and cultural LS) is needed. For this second section, we developed an instrument of 14 items to assess users' perceptions of LS quality as provided by COS, according to Larson et al.'s questionnaire [33]. The answers were rated on a five-point Likert scale (1—very poor, 2—poor, 3—neutral, 4—rich, and 5—excellent) [33].

Finally, Section 3 aims to identify the users' profile by defining the following characteristics: age, sex, occupation, branch of knowledge, time spent at UPV, and frequency of COS use. According to COS frequency, university community members were classified as low, medium, and high users. Low users are participants who never use COS, or only once a week, medium users two times a week, and high users three, four, or more than four times a week.

### 2.5. Data Analysis

The questionnaire was available for one month (October 2019). Data were automatically downloaded as a Microsoft Excel file (.xlsx). Statistics tests were performed with two statistics software solutions: SPSS Statistics 23 (IBM, Armonk, NY, USA) and Statgraphics Centurion XVII (Statgraphics Technologies, The Plains, VA, USA). Descriptive analyses were carried out. Basic statistics and measures of normality, symmetry, and kurtosis were obtained for the information related to the level of satisfaction, needs, perceptions of landscape services' quality, and types of LS.

A binary system was used to mark the LS identification (1 = yes; 0 = no) to assess LS provided by campus open spaces and the needs related to COS management. As our data was not normally distributed, non-parametric tests were used to find significant differences between the university community's answers.

Significant differences of the perceived number of well-provided LS and COS preferences with different UGI elements were calculated with the Kruskal–Wallis test between groups of users according to age, occupation, time spent at UPV, and frequency of COS use.

The Kruskal–Wallis test is a rank-based non-parametric test that can be used to determine if there are statistically significant differences between two or more groups in situations where the dependent variable is measured at least at an ordinal level, or when the dependent variable is measured at an interval level. It is used when the assumptions of one-way ANOVA are not met [34].

A Mann–Whitney U test or Mann–Whitney–Wilcoxon (MWW) test was used for the sex and branch of knowledge. Mann–Whitney–Wilcoxon tests for equality of location are applicable only when the two underlying distributions have an equal shape, and this assumption is difficult to verify in practice. The null hypothesis (H0) of the Mann–Whitney U test (Wilcoxon's two-sample test or Wilcoxon's rank sum test) is accepted when the observations of the two samples are from the same distribution. The alternative hypothesis (H1) states that the observations of the two samples are from two distributions that have the same shape, but there is a shift in location [35].

Bonferroni's correction was between groups of users to counteract the problem of multiple comparisons with a 95.0% confidence level ($p < 0.05$) [36].

The bivariate Pearson correlation is a parametric measure commonly used to measure correlations between pairs of variables. Pearson's correlation matrix produces a sample correlation coefficient, *r*, that measures the strength and direction of linear relationships between pairs of continuous variables [37]. Pearson's correlation matrix was used to observe differences among the following variables: age, occupation, education level, branch of knowledge, and time spent at UPV. Pearson's correlation matrix was built for items related to atrium, garden, park, and green verge.

## 3. Results

### 3.1. Sample Description

A total of 828 persons filled out the 61-question survey. Only those questionnaires that were at least 93% completed were accepted. The sample consisted of 786 participants, of whom 49.4% were women, 46.8% were men, and 3.8% of participants chose not to answer. Respondents' ages ranged from 18 to over 50 years, but the largest faction (35.6%) were 18 to 22 (Table 3).

**Table 3.** Categories percentage values from the sample distribution.

| Age | | Time Spent at UPV | |
|---|---|---|---|
| 18–22 | 35.6 | One year and less than one year | 15.8 |
| 22–30 | 18.8 | Two–five years | 23.6 |
| 30–50 | 23.1 | Between six and ten years | 13.0 |
| >50 | 18.7 | More than ten years | 44.5 |
| No data | 3.8 | No data | 3.1 |
| **Occupation** | | **Frequency of COS use** | |
| Student | 60.9 | Low users | 21.9 |
| AdSS | 20.2 | Medium users | 16.3 |
| TRS | 17.7 | High users | 60.7 |
| Other | 1.2 | No data | 1.1 |
| **COS preference** | | **Level of studies** | |
| Open space | 74.6 | Secondary school/High school | 31.4 |
| Indoor space | 8.0 | Vocational education | 4.7 |
| Sport space | 11.6 | Undergraduate | 20.7 |
| No data | 5.8 | Master's | 22.5 |
| | | Doctorate | 19.9 |
| | | Other | 0.8 |
| **Knowledge branch** | | **Sex** | |
| Landscape-related disciplines | 39.1 | Woman | 49.4 |
| Other disciplines | 59.7 | Man | 46.8 |
| No data | 1.2 | I prefer not to answer | 3.8 |

Of the participants, 60.9% were students (undergraduate or graduate), 20.2% were administration and services staff (AdSS), and the rest (17.7%) teaching and research staff (TRS). Within AdSS, the majority were respondents over 30 years old (43.3%) and over 50 years old (44.5%). Around 50% of TRS were over 50. Their relationship with the university is conditioned by time, so a large part of the university community had been working at the UPV campus for more than ten years (44.5%), while first-year students made up 15.8%. As for their highest degree earned, 31.4% responded secondary school or high school, 4.7% vocational school, 20.7% undergraduate, 22.5% master's, and 19.9% were already doctors. Finally, 308 persons (39.1%) came from landscape-related disciplines (LRD) (e.g., forestry, agriculture, environmental engineering, architecture and fine arts), and 470 persons (59.7%) belonged to other disciplines (OD) (e.g., economics, computer science, civil engineering, etc.). The connection of the first group of disciplines with landscape through its analysis, planning, intervention, or representation was assumed to involve a higher sensitivity to the environment and open space. According to COS frequency, high users is the main category which engages with COS three, four, and more than four times a week (60.7%).

Positive significant correlations ($p < 0.01$) (bold numbers in Table 4) were observed between the following variables: age, occupation, level of studies, and time spent at UPV. Negative, but weaker, significant correlations were observed between the age, occupation, level of studies, branch of knowledge, and time spent at UPV parameters.

**Table 4.** Pearson correlation matrix between respondents' characteristics.

| Correlations | Age | Sex | Occupation | Level of Studies | Knowledge Branch | Time Spent at UPV | Frequency |
|---|---|---|---|---|---|---|---|
| Age | 1 | 0.080 * | **0.719 **** | **0.722 **** | −0.054 | **0.756 **** | −0.124 ** |
| Sex | | 1 | 0.128 ** | 0.063 | 0.181 ** | 0.122 ** | −0.080 * |
| Occupation | | | 1 | **0.650 **** | −0.012 | **0.662 **** | −0.185 ** |
| Level of studies | | | | 1 | −0.003 | **0.667 **** | −0.187 ** |
| Knowledge branch | | | | | 1 | 0.040 | −0.121 ** |
| Time spent at UPV | | | | | | 1 | −0.114 ** |
| Frequency | | | | | | | 1 |

* Correlation is significant at the 0.05 level; ** Correlation is significant at the 0.01 level (2-tailed); bold is used to highlight values higher than 0.6.

### 3.2. Level of Satisfaction

Regarding the satisfaction with the general state and management of COS, almost 33% of participants were unsatisfied or had a neutral opinion, while 33.19% were satisfied (Table 5). Within the groups of COS users, only four of them expressed a higher percentage of satisfaction with the COS' current management, eight showed a negative appreciation, and six were neutral. The respondents with the highest perception of outdoor areas were university community members over 50, AdSS staff, women, and high users (Table 5). Significant differences (H = 18.1; $p$ = 0.000 *) were found between first-year students and respondent groups who had been working or studying at UPV for more than six years (Table 5, bold numbers). There is no clear relationship between the time spent at UPV and general satisfaction with COS, but people who spend more time on campus and get to know it better are more critical than the other groups. Surprisingly, for users who have been at the UPV campus for more than 10 years, the satisfaction level is higher.

**Table 5.** University community's degree of satisfaction with COS' condition and management.

| Variables | Participants | Good (%) | Neutral (%) | Bad (%) |
|---|---|---|---|---|
| | University community | 33.19 | 33.87 | 32.92 |
| | Groups of users | | | |
| Age | 18–22 | 32.97 | 34.43 | 32.6 |
| | 22–30 | 27.97 | 36.36 | 35.66 |
| | 30–50 | 33.53 | 32.37 | 34.10 |
| | >50 | 39.84 | 29.69 | 30.47 |
| Occupation | Student | 31.40 | 33.76 | 34.84 |
| | AdSS | 41.89 | 32.43 | 25.68 |
| | TRS | 28.81 | 35.59 | 35.59 |
| Branch knowledge | Landscape-related disciplines | 33.79 | 32.08 | 34.13 |
| | Other disciplines | 32.95 | 34.77 | 32.27 |
| Time spent at UPV | One year and less than one year | 36.67 | 37.5 | 25.83 |
| | Two–five years | 32.40 | 34.08 | 33.52 |
| | Between six and ten years | 31.68 | 29.70 | 38.61 |
| | More than ten years | 31.65 | 35.44 | 32.91 |
| Sex | Women | 29.03 | 32.9 | 38.06 |
| | Men | 27.64 | 39.02 | 33.33 |
| Frequency | Low users | 36.64 | 32.45 | 30.91 |
| | Medium users | 27.6 | 39.0 | 33.3 |
| | High users | 36.6 | 32.5 | 30.9 |

*3.3. Campus Open Space Needs*

An average of five needs were observed. More than 50% declared that natural shade, water, trees, tables, and benches were the main deficiencies related to COS (Figure 4).

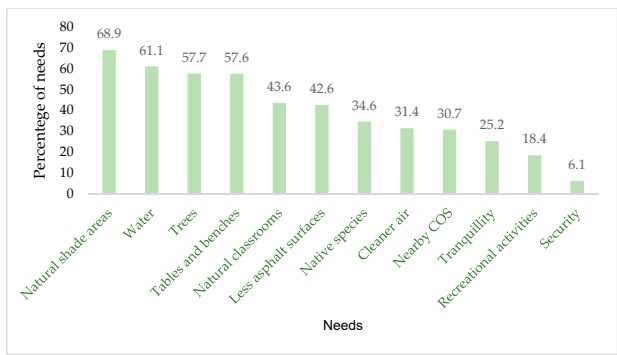

**Figure 4.** Percentage of needs identified by university community regarding the current state and management of COS.

Users' top five needs could be divided into three environmental needs (natural shade, water, and trees, with an average of 62.6%) and two functional needs (tables and benches and using COS as natural classrooms, with an average of 50.6%) (Figure 4). All groups expressed between three and four needs with a high percentage (>50%) (Table A1, Appendix A).

According to the user profile, the university community expressed different priorities. There were significant differences between women and men and users from different branches of knowledge (Table 6). Women wanted to increase the number of urban facilities more than men do (Mean Median Range: 410.6 (women) and 345.6 (men)) and also use COS as natural classrooms (Mean Median Range: 404.3 (women) and 352.3 (men)). On the contrary, for men, tranquillity, cleaner air, and more trees were perceived as better (Table 6). According to the branch of knowledge, two significant differences between users with landscape-related disciplines and other disciplines were found (Table 6). Users with a landscape focus expressed that more native species need to be introduced and that the COS needs to be used as a learning environment.

**Table 6.** Significant differences between the needs identified by women and men, and respondents with landscape-related (LRD) and other disciplines (OD); U Mann–Whitney non-parametric test (W), *p*-values and Mean Median Range.

| Needs | W | *p* | Women | Men |
|---|---|---|---|---|
| Tables and benches | 59,298.0 | 0.000 * | 410.6 | 345.6 |
| Tranquillity | 81,074.0 | 0.000 * | 354.6 | 404.8 |
| Cleaner air | 76,784.0 | 0.030 * | 365.6 | 392.2 |
| Trees | 77,455.5 | 0.022 * | 363.9 | 395.0 |
| Natural classrooms | 61,748.5 | 0.000 * | 404.3 | 352.3 |
| **Needs** | **W** | ***p*** | **LRD** | **OD** |
| Native species | 67,045.0 | 0.034 * | 406.8 | 378.1 |
| Natural classrooms | 62,889.0 | 0.000 * | 420.3 | 369.3 |

* Significant differences (*p* < 0.05).

*3.4. COS' Current State and Management Given by the Quality of Landscape Services Provided*

University community members perceived eight LS provided by COS as having good quality, with values over 50%. In descending order, they were: get together with friends or meet up, walks, relax, temperature/light, air quality (AIR), sport, daily basic needs (DAILY NEEDS), and art creation (ART) (Figure 5).

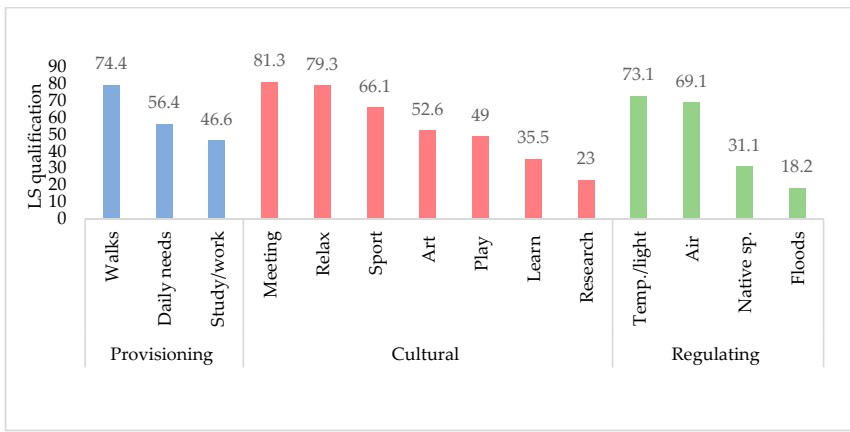

**Figure 5.** Participants' perception (expressed by %) of good quality landscape services (LS) provided by COS.

In terms of age, the youngest participants had in general a better opinion of the quality of all LS than the rest of the university community (Table A2, Appendix B). The first-year students and COS high users were the least demanding, deeming eleven LS to be of very good quality, while respondents who had spent more than ten years at UPV and low users registered around seven complaints, which were related to the current state of the following LS: study/work, research, learn about environment, protection floods, and native species (Appendix B).

Significant differences ($p < 0.05$) were calculated for the groups of users according to age, occupation, time spent at UPV, and frequency of COS use. Table 7 presents the landscape services where significant differences were observed. The last three columns show that the most frequent pairs of groups were the 18–22-year-old and over 50 respondents, students and teachers, and participants who had spent a short and a long time at the UPV campus.

**Table 7.** Significant differences between respondents' perceptions of LS provided with good quality, according to user profile. Kruskal–Wallis non-parametric test (H), *p*-values and significant pairs of groups.

|  | LS | H | *p* | Groups | | |
|---|---|---|---|---|---|---|
| | Walks | 12.6 | 0.005 * | 18–22 & 22–30 | | |
| | Study/work | 16.1 | 0.001 * | 18–22 & 22–30 | 18–22 & >50 | |
| | Daily needs | 19.6 | 0.000 * | 18–22 & >50 | | |
| Age | Play | 12.7 | 0.005 * | 18–22 & >50 | | |
| | Research | 17.7 | 0.000 * | 18–22 & >50 | | |
| | Art | 16.0 | 0.001 * | 18–22 & 30–50 | 18–22 & >50 | |
| | Temp./light | 13.7 | 0.003 * | 18–22 & 22–30 | 18–22 & 30–50 | |
| Occupation | Play | 7.7 | 0.021 * | Student & TRS | | |
| | Art | 13.7 | 0.001 * | Student & TRS | | |
| | Daily needs | 11.2 | 0.010 * | ≤1 & >10 | | |
| | Play | 23.7 | 0.000 * | ≤1 & >10 | 2–5 & >10 | |
| Time spent | Sport | 13.8 | 0.003 * | ≤1 & >10 | | |
| at UPV | Learn | 18.8 | 0.000 * | ≤1 & 2–5 | ≤1 & 6–10 | ≤1 & >10 |
| | Art | 22.5 | 0.000 * | ≤1 & >10 | | |
| | Air | 12.2 | 0.006 * | ≤1 & >10 | | |
| | Temp./light | 11.8 | 0.008 * | ≤1 & >10 | | |
| | Research | 9.8 | 0.007 * | Low & High | | |
| Frequency | Learn | 6.8 | 0.031 * | Low & High | | |
| | Art | 14.8 | 0.000 * | Low & Medium | Low & High | |

\* Significant differences ($p < 0.05$).

The U Mann–Whitney test (W) displayed significant differences between women and men (Table 8, Appendix B). Women perceived a higher quality of all LS displayed in Table 8 than men, demonstrated by higher values of Mean Median Range associated with women (last two columns).

**Table 8.** Significant differences of the perceptions regarding the quality with which COS provide LS between women and men. U Mann–Whitney non-parametric test (W), *p*-values and Mean Median Range.

| LS | W | *p* | Women | Men |
|---|---|---|---|---|
| Research | 51,174.0 | 0.000 * | 371.8 | 320.9 |
| Learn | 53,328 | 0.000 * | 380.6 | 327.4 |
| Art | 55,144.5 | 0.001 * | 378.5 | 332.7 |
| Meeting | 59,307.5 | 0.013 * | 380.0 | 345.0 |
| Floods | 53,414.5 | 0.040 * | 356.8 | 327.4 |
| Native species | 57,189.0 | 0.042 * | 368.9 | 338.7 |

\* Significant differences (*p* < 0.05).

### 3.5. Preferences of University Community Regarding COS Typology

The favourite COS were garden and park (96.9% and 89.5%) (Table 9). These outdoor areas achieved the highest percentage of agreement with the six descriptive attributes used in the questionnaire. More than 80% of users considered park and garden to be beautiful, natural, maintained, cheerful, useful for the management of environmental functions, and representative of the image of COS (Table 9).

The differences of agreement between the least green elements (atrium and green verge) and the greenest components (campus, park) are very evident for almost all items, excepting the "maintained" item, which had higher scores (more than 60%) than the rest of the items.

**Table 9.** Percentage of the university community's preferences of selected COS and percentage of associated attributes.

| COS | Attributes and Preferences (%) | | | | | | |
|---|---|---|---|---|---|---|---|
| | Like | Beautiful | Natural | Maintained | Cheerful | Environ Functions | Campus Image |
| Garden * | 96.9 | 89.4 | 89.9 | 88.3 | 85.8 | 89.6 | 87.3 |
| Park * | 89.5 | 84.6 | 86.0 | 87.7 | 80.9 | 85.8 | 82.7 |
| Green verge | 32.4 | 38.2 | 23.7 | 60.9 | 30.0 | 33.7 | 48.0 |
| Atrium | 11.6 | 19.8 | 3.3 | 66.9 | 7.4 | 7.4 | 26.8 |

\* The most popular and greenest COS.

The classification of preferences between groups of users according to the six variables from Table 10 is similar to the general situation of users' opinions regarding all descriptive COS items: garden and park hold the first two places, followed by green verge (GV), and the lowest score is for atrium.

The preferences of university community groups are more homogeneous for garden, park, and green verge than for atrium, where significant differences were calculated. The highest appreciation for garden (98.11%) and park (92.16%) was expressed by high COS users, while the new arrivals liked green verge (39.52%) and atrium (19.35%) the most (Table 10). Between the preferences of respondents for atrium, there were statistically significant differences (*p* < 0.05) by age, occupation, and branch of knowledge. The highest percentage of agreement was expressed by those who have experienced the UPV for a short time (19.35%), the youngest users (18–22 years) (17.20%), and students (14.23%) (Table 10). The worst appreciation for atrium (6.86%) (Table 10) was given by the university community members who had spent more than ten years on campus. For park, there were significant differences between participants aged 22–30, 30–50, and over 50 (Table 11). The highest percentage of respondents who liked park was expressed by users with 30–50 years old (91.76%) (Table 10) and the lowest by the

younger group (22–30) (85.71%). Regarding the branch of knowledge, there were significant differences among users, as 10.71% of respondents with LRD and 12.37% with OD liked atrium (Table 10%). In Table 11 the Mean Median Range was calculated for both groups: 356.4 landscape-related disciplines and 405.5 other disciplines.

**Table 10.** Percentage of group respondents who liked COS.

| Variables | Respondents | Garden | Park | GV | Atrium |
|---|---|---|---|---|---|
| Age | 18–22 years | 96.07 | 88.89 | 36.20 | 17.20 |
|  | 22–30 years | 97.30 | 85.71 | 25.00 | 8.78 |
|  | 30–50 years | 98.90 | 91.76 | 33.15 | 8.79 |
|  | >50 years | 95.24 | 89.80 | 31.97 | 8.84 |
| Sex | Woman | 97.69 | 90.72 | 35.66 | 12.60 |
|  | Man | 96.47 | 88.01 | 29.89 | 10.90 |
| Occupation | Student | 96.87 | 88.26 | 32.08 | 14.23 |
|  | AdSS | 97.48 | 92.45 | 31.45 | 6.92 |
|  | TRS | 96.40 | 90.65 | 33.09 | 7.91 |
| Branch of knowledge | Landscape-related disciplines | 95.45 | 87.30 | 30.72 | 10.71 |
|  | Other disciplines | 97.87 | 90.83 | 33.19 | 12.37 |
| Frequency of COS | Low users | 95.35 | 84.88 | 30.23 | 13.37 |
|  | Medium users | 94.57 | 86.82 | 27.13 | 12.40 |
|  | High users | 98.11 | 92.00 | 34.74 | 10.92 |
| Time spent at UPV | One year and less than one year | 95.16 | 88.71 | 39.52 | 19.35 |
|  | Two–five years | 95.70 | 85.95 | 36.76 | 17.30 |
|  | Between six and ten years | 98.04 | 92.16 | 28.43 | 9.80 |
|  | More than ten years | 98.00 | 90.83 | 29.51 | 6.86 |

**Table 11.** Significant differences between respondents' preferences for atrium and park according to age, occupation, and branch of knowledge. Kruskal–Wallis non-parametric test (H), U Mann–Whitney non-parametric test (W), $p$-values, and significant pair of groups.

| Variable | Like | H | $p$ | Groups (Kruskal–Wallis Test) | | |
|---|---|---|---|---|---|---|
| Age | Atrium | 28.0 | 0.000 * | 18–22 & 22–30 | 18–22 & 30–50 | 18–22 & >50 |
|  | Park | 16.5 | 0.000 * | 22–30 & 30–50 | 22–30 & >50 |  |
| Occupation | Atrium | 16.5 | 0.000 * | Student & AdSS | Student & TRS |  |
|  |  |  |  | Groups (U Mann–Whitney test) | | |
|  |  | W | $p$ | LRD | OD |  |
| Branch of knowledge | Atrium | 80,197 | 0.002 * | 356.4 | 405.5 |  |

\* Significant differences ($p < 0.05$).

### 3.6. Benefits Provided by COS with Two Different UGI Elements (Park and Green Verge)

For green verge, an average of four landscape services was identified. The most perceived LS was pass through (87.8%), followed by social interaction (46.5%) and walk/do sport (40.2%) (Table 12). The three LS themes (provisioning, regulation, and cultural) were poorly represented (Table A3, Appendix C). All users identified pass through as the main LS, but few filled in a second LS. More than 50% of young new students and women perceived social interactions as a benefit provided not only by park but also by green verge.

For park, an average of nine landscape services was identified. The most perceived LS was relaxing (93.0%) and other twelve LS were mentioned often (50–90%) (Table 12, Table A4 (Appendix D)). All LS themes were well represented: 100% provisioning LS, 88% cultural LS, and 75% regulation LS (Appendix D).

**Table 12.** University community's percentage of perceived benefits.

| Theme | Landscape Services | Green Verge | Park |
|---|---|---|---|
| Provisioning | Pass through | 87.8 | 49.8 |
| | Work/study | 9.8 | 58.4 |
| | Eat | 28.2 | 80.2 |
| Cultural | Relax | 31.1 | 93.0 |
| | Play | 7.9 | 67.1 |
| | Walk/do sport | 40.2 | 61.6 |
| | Education | 11.7 | 56.0 |
| | Research | 5.8 | 28.6 |
| | Inspiration | 19.4 | 59.3 |
| | Psychological connections | 15.9 | 50.6 |
| | Meeting | 46.5 | 66.1 |
| Regulation and maintenance | Flood control | 5.5 | 21.2 |
| | Air quality | 17.5 | 78.5 |
| | Climatic conditions | 12.5 | 64.2 |
| | Biodiversity | 10.7 | 58.1 |

Significant differences ($p > 0.05$) in the type of LS provided by green verge and park were found between men and women (Table 13). Women perceive a higher potential for those outdoor areas to provide all types of LS than men, e.g., the Mean Median Rank of perceived biodiversity for green verge was 390.5 (women) and 366.8 (men) and for park was 395.9 (women) and 361.1 (men) (Table 13). Significant differences (W: 66984.0; $p = 0.039$ *) (Table 13) were found between respondents from different branches of knowledge. More students with a focus on landscape-related disciplines (Mean Median Rank: 407.0) thought park provides an outdoor place to work and study than respondents with other disciplines (Mean Median Rank: 378.0).

**Table 13.** Significant differences of perceived LS between women (Wom) and men and respondents with different branches of knowledge (Landscape-related (LRD) and other disciplines (OD)). U Mann–Whitney non-parametric test (W) and *p*-value.

| LS | Green Verge | | | | Park | | | |
|---|---|---|---|---|---|---|---|---|
| | W | *p* | Wom | Men | W | *p* | Wom | Men |
| Work/Study | - | - | - | - | 60,864.5 | 0.000 * | 406.5 | 349.9 |
| Eat | 65,918.0 | 0.016 * | 393.5 | 363.6 | 63,423.5 | 0.000 * | 400.0 | 356.8 |
| Play | - | - | - | - | 60,422.5 | 0.000 * | 407.7 | 348.7 |
| Walk/Sport | 66,001.5 | 0.029 * | 393.3 | 363.9 | - | - | - | - |
| Education | 67,690.5 | 0.020 * | 389.0 | 368.4 | 65,096.5 | 0.012 * | 395.7 | 361.4 |
| Research | 68,411.0 | 0.010 * | 387.1 | 370.4 | 63,074.0 | 0.000 * | 400.9 | 355.9 |
| Inspiration | - | - | - | - | 62,594.0 | 0.000 * | 402.1 | 354.6 |
| Psych. connection | - | - | - | - | 64,129.5 | 0.004 * | 398.1 | 358.8 |
| Meeting | 64,750.0 | 0.008 * | 396.5 | 360.5 | 62,662.0 | 0.000 * | 401.9 | 354.8 |
| Air quality | 66,223.5 | 0.007 * | 392.8 | 364.5 | 65,999.5 | 0.008 * | 393.3 | 363.8 |
| Climatic | - | - | - | - | 63,529.5 | 0.001 * | 399.7 | 357.1 |
| Biodiversity | 67,096.5 | 0.006 * | 390.5 | 366.8 | 64,996.5 | 0.010 * | 395.9 | 361.1 |
| | | | | | W | *p* | LRD | OD |
| Work/Study | | | | | 66,984.0 | 0.039 * | 407.0 | 378.0 |

* Significant differences ($p < 0.05$).

## 4. Discussion

*4.1. Design and General Perceptions of UPV's Campus Open Spaces*

Results show the ability of the UPV's COS to provide a large range of good quality landscape services in terms of user perception, despite the lack of consensus in the level of satisfaction with the COS' current state and management.

As expected, meeting with friends, relaxing, and providing a place to pass through are the main functions of the campus open spaces, perceived as such by almost 80% of respondents. Our results are consistent with Ulrich et al.'s findings [38], who affirmed that stress reduction and increasing social relationships are the most important purposes of open spaces. The ability of open spaces to satisfy daily basic needs is a highly rated landscape service. At lunchtime, many students sit on the grass eating, resting, or even taking a nap [1,12].

These principal functions perceived by the university community are related to provisioning and cultural landscape services. We speculate that users perceive these two LS categories, rather than the LRD, which is poorly perceived.

Cultural landscape services contribute to different purposes: health/enjoyment, self-fulfilment (personal), and social fulfilment. Two pairs of LS can be distinguished: active enjoyment and physical health and passive enjoyment and mental health. Leisure activities (such as sport, walking, running, or playing in open places) are directly related to increasing physical health. Considering daily activity classes, it was found that "pass through" is an important benefit which was not included in any existing category of ecosystem service classifications [8].

Most of the respondents agree with the open space landscape services. However, there are significant differences between user profiles concerning the number of good quality landscape services identified. In particular, respondents over 50 tend to rate the landscape services provided by UPV's COS lower than the youngest respondents. People who have spent more than half of their life at UPV (thirty years) rate half of LS poorly and, surprisingly, indicate the smallest number of needs (3 of 12). We speculated that, on the one hand, age is connected to a longer relationship with the university and more experience, which probably means that users over 50 are perceived as being more thorough than younger people and also as having another vision of UPV values. The majority of young users have recently come into contact with COS (34.1%) or have spent between two and five years there (49.1%). On the other hand, there are some factors, such as schedules, different daily concerns, and urban furniture, which might influence the respondents' decision on how to use campus open spaces. Design elements in outdoor areas (e.g., benches, tables, lawn, and natural shade) have the potential to provide short-term uses, which attract the interest of the university community. In addition, daily habits and higher frequency of COS use on the part of students means they know those open spaces better and will create bonds with the natural environment. This is reflected in the students' high rate of positive responses about the general quality of LS provided by COS.

Regarding environmental benefits, around 70% of participants rated the provision of adequate temperature and light highly. However, the analysis of user needs suggests that people are demanding that this service be improved, since natural shade and trees are among the most requested needs. This landscape service is especially important in the Mediterranean context, characterised by high temperatures, insolation, and scarce precipitation. The pleasant weather in the Mediterranean region has a notable influence on the use of outdoor areas [12,39]. At the UPV, sunny and warm days have a major impact, especially on foreign exchange students, who intensively enjoy the outdoors, and sometimes even more so than local students, spending hours sitting on the lawn, listening to music, talking on the phone, resting, reading, studying, or hanging out with friends.

Regarding the frequency of COS use, there are no significant differences between women and men (respectively, 51.6% and 48.4%), which is not consistent with Speake et al. [9], who found out that men spent more time outdoors on campus than women.

With regard to users' needs, respondents seem satisfied with the COS' ability to offer protection, safety, and tranquillity, and their demands are mainly related to green space composition. Accordingly, they ask for more "green-blue" elements, such as more natural shade, more trees, and more water. But they also ask for more green space furniture like tables and benches. User demands suggest that the utilitarian and ornamental view of green space is dominant over the ecological view. However, two things are remarkable: more than 40% of users are against asphalted surfaces, and almost 35% of respondents request more native species. In this way, the UPV's COS is lacking an ecological design. Green elements have been added over the course of time to cover residual spaces. The UPV's campus open spaces' history responds to the complaints by the university community about the number and the distribution of campus open spaces, as was reflected by questionnaire answers.

The UPV's campus was built in three phases, starting in 1970 with the Higher Technical School of Building Engineering and ending with the Polytechnic City of Innovation in the 1990s. There was no initial sustainable campus open space design [18], because a variety of factors, such as accessibility, being close to school, and increasing the number of schools, were the main priorities for university members at that time, more than building a natural campus ecosystem. At the beginning, the UPV's campus open space went through functional remodelling at each stage, followed by the creation of a green axis made up of a large central park, whose purposes were to facilitate crossing the campus (pass through LS), guide the pedestrian flow, and improve the connectivity of the common areas with campus open spaces associated with each school. In the last building phase, a number of non-native species were introduced with the intention of creating an arboretum of exotic trees. However, vegetation has been introduced in a dispersed way to increase the diversity of tree species [23], and the UPV campus has few areas which mimic a natural forest structure.

The design of the UPV's campus open spaces is in line with Cooper and Wischemann's campus outdoor areas design recommendations [1], which facilitate different types of uses depending on the location of the campus open space, e.g., the front lawn is seen as a home base, which invites students to sunbathe, meditate, and relax (actions they do not want to engage in in other public spaces), while the central plaza, besides its directional goals, provides a place where friends can meet, walk, have lunch, and watch and participate in different school activities. Pedestrian circulation is part of the COS users' life and is provided by the design of the UPV campus' ordered streets. Well-designed campus open spaces should convey coherence, clarity, and comprehensibility [14].

The provision of connectivity between outdoor areas is a key aspect, and future improvement actions should go beyond functional criteria. It is important to provide connectivity between outdoor areas not only physically, but also at the ecological level, which entails the involvement of other complex processes. In this regard, the Green Flag recommends the role of urban green spaces in enhancing ecological networks of habitats and species populations. These principles have been integrated into the UPV's green campus guidelines [6], which include strategies to increase native species and enhance biodiversity.

*4.2. Greener or Greyer COS Community Preferences*

4.2.1. Campus Open Space General Preferences

Results highlight the role of green areas in users' choice and evaluation. The analysis of preferences for specific types of urban green infrastructure reveals the university community's preferences for greener campus open spaces (park, garden). These green infrastructures contain a high percentage of permeable cover given by vegetation (tree canopy, shrubs, and lawn) or permeable tracks, high numbers for tree biodiversity and species diversity, and complex habitat heterogeneity (number of vegetation strata). A big difference is also outlined between the greener (garden, park) and greyer (atrium, green verge) COS when asking about the value for the university image. Users perceived that precisely those COS characterised by a high level of greenery are the most valuable elements in the UPV's public image.

The preferred COS are garden and park. A high level of agreement regarding associated positive attributes, e.g., beautiful, cheerful, natural, useful for environmental functions, and maintained, are associated with greener COS. In this sense, the garden which is located next to the Higher Technical School of Architecture has a large permeable green area (98%) with a wide array of tree species which provide shade and a small pond that adds a cool feeling and a sense of tranquillity. Different sculptures and benches are spread across the lawn, and one narrow paved road facilitates crossing the area. In addition, the park is located at the end of the campus' central green area and is surrounded by different buildings. It is characterised by a large green cover area of lawn and trees (96%) with scattered sculptures and historical pieces. There are many studies which reveal people's preference for greener landscapes [9,10,12,40]. Generally, people are looking for open spaces which can provide positive thoughts and emotions, e.g., relaxation, peace, happiness, and joy [41].

On the opposite end of the spectrum, green verge and atrium are the least preferred. Green verge is a green element surrounded by paved campus streets (60%), which contains only two species of trees (8%). This COS includes urban furniture. Nearby, there are some facilities, such as a swimming pool, a tennis court, and the Casa del Alumno (the student centre). This building is surrounded by green areas and artistic and scientific creations, part of the Campus Escultòric Museum. Atrium, located in front of the Higher Technical School of Agronomic and Natural Environment Engineering, consists of a large hard-surface area (92% paved space) with trees planted in pots above an underground parking which provide hardly any natural shade or biodiversity.

Concerning the atrium, it is remarkable that the value assigned by younger (students) participants was slightly higher than the value assigned by older ones (AdSS and TRS). This fact may be explained by the way students use the space. The atrium, which corresponds to the concept of backyard [1,12] regarding its functions and location in the campus, seems more attractive to students. It is a partially enclosed courtyard with meeting elements which can be used in a temporary fashion, e.g., to have lunch sitting or leaning on paved plant pots because "people feel more comfortable with a wall or plant on their back" [1]. The defining characteristic of this place is its privacy. Students come here alone or in pairs to talk, relax, or study because they feel a greater sense of privacy than on the front yard (garden) [12].

4.2.2. Perception of Landscape Services Provided by Green Verge Versus Park

Results show the existence of differences between two COS (park and green verge) with different permeable cover in terms of their perceived landscape services. These differences could be related to their level of multifunctionality. A more complex and natural COS with large and permeable areas, like the park, is perceived as having a higher potential for providing more and different landscape services than a COS poor in green areas, even with urban furnishing. In addition, results reflect the relationship between respondents' preferences and the landscape services provided by specific types of urban green infrastructure.

Park is one of respondents' favourite and most appreciated outdoor spaces, and it is perceived as providing an average of nine benefits (13 out of 15 LS make up more than 50% of agreement among users) connected to all three themes (provisioning, cultural, and regulating). A high percentage of LS identified by the university community's preferences and its function connecting diverse student facilities (swimming pool, Casa del Alumno, football pitch, car parks, and coffee shops) make park a well-known area used by everyone. On the other hand, for green verge there is only one landscape service, "pass through", which is identified by more than 50% of respondents (87.8%). Despite its popular location (next to the Student Centre), green verge is more a pass-through open space in the middle of a crowded intersection because of the pedestrian flow.

Generally, COS are perceived as suitable outdoor places for social interaction, where casual meetings between university community members occur, and spaces to practice sport, which is consistent with Cooper and Wischemann's results [1]. However, park is perceived as more adequate

for gatherings with friends and walking, which supports the findings of Lau et al. [14], according to which more spacious open spaces accommodate transitional and social activities better.

Links are developed between users and COS which are assessed according to the multiple use of outdoor areas. A larger percentage of users (49.3%) perceived park as an open space, which encourages the development of psychological and cultural connections between university community members. This result agrees with Cooper and Wischemann's theory [1], which says that COS inspire strong feelings of belonging and a motivation to come back to fulfil daily tasks. A high frequency of COS use and a long relationship with the UPV's campus make the university community more familiar with the campus background and its facilities. According to Cooper and Wischemann [1], COS seems to be an "outside world" which differentiates between private life and peoples' job obligations.

The provision of relax and stress reduction (93%) is mainly related to park. Green "outside" spaces are mostly associated with calm, relaxation, serenity, peace [14], and university campus restorativeness [11].

Respondents' perceptions of park as a more natural outdoor space, more useful for environmental functions than atrium or green verge, support prior findings about perceived greenness more associated with central campus green areas [11]. Park is covered by a large expanse of lawn, a less natural type of cover, with high maintenance which requires substantial quantities of water and an efficient irrigation system. UPV's Environmental Unit [6] considered it an expensive COS management method and maybe not the best green strategy for the entire campus. Lau et al. [14] recommend maintained or grown lawn areas because their "attractive sight of lush green is largely welcomed by people". Lawn areas are ideal as playing surfaces [14], especially for summer school community members. Because people of all ages use these outdoor areas often and intensively, the wasteful irrigation aspect should be taken into consideration to provide proper functional conditions and improve the design of the UPV's campus.

Unlike the students of Liverpool's Hope Park campus [9], who preferred gardening over natural spaces, more than half of UPV students identify, besides social and functional reasons to use COS, their related environmental benefits (66.6% for park and 12.7% for green verge). Even if the campus of the Universitat Politècnica de València does not have truly natural areas (forests), for the university community the current green areas are very important. Unlike Liverpool's respondents [9], the UPV's COS users from landscape-related disciplines demonstrate awareness of, and care about, natural issues, in terms of introducing more native species and giving an environmental–didactic use to these natural laboratories.

## 5. Limitations of the Study

Limitations of the study include possible bias traditionally introduced by online surveys, which could lead to under- or over-representation of different cohorts. As discussed above in this paper, relations of causality could be affected by socio-demographic characteristics (for instance age or the time spent at UPV).

This work seeks to know the benefits perceived by users of open spaces, focusing on two of the most popular ones. Future research will address more specific aspects for improving open spaces, e.g., the expected/preferred ratio between green space and urban space. Qualitative techniques, such as in-depth interviews and focus groups could be added in order to supplement quantitative methods.

## 6. Future Research

Further quantitative analyses will be carried out in the future to assess high-quality benefits and target users' perceptions and preferences, which will be useful for landscape planners.

As stated above, a large amount of information was collected for this study. Future research will consider additional variables, such as viewing from adjacent university buildings and significant places for university community. Furthermore, our study was unable to demonstrate significant differences

in COS needs between women and men, but we suggest further studies of the university community's demands according to sex by campus landscape planners.

Regarding the analysis methodology, Structural Equation Modelling will be used to explore the interrelationships among the variables of the study simultaneously, considering the possible correlations observed.

## 7. Conclusions

This article reviews the university community's perceptions of the general situation of the campus' outdoor environment. A web-based survey was carried out in the UPV with all of the campus' community members. Statements connected to the three principal themes of landscape services were used to evaluate the perceived quality of landscape services provided by campus open spaces. The university community rated the cultural and provisioning benefits highly due to their daily interactions with the campus environment.

A survey including four examples of outdoor areas with different urban green infrastructure elements revealed that the university community is attracted to campus open spaces with more extensive green surfaces (gardens and parks). However, a greener campus does not necessarily mean only enlarged lawns, but a more complex landscape, where human actions are in symbiosis with the natural ecosystem.

Open spaces with abundant green cover are perceived as beautiful, cheerful, useful for the maintenance of environmental functions areas, and suitable to the UPV's campus' image. The youngest students also appreciate the more paved surfaces (atrium and green verge) for their social function, since they are perceived as meeting points within a more off-the-beaten-track environment, where daily casual encounters happen.

For a green campus, green criteria, eco-friendly infrastructure, and effective management of campus open spaces must all be included. A well-designed campus open space with functional, environmental, or psychological elements creates the perception of a multifunctional place: a learning environment, a private environment for social interaction, a natural environment to enjoy nature, and a meditative environment to disconnect and relax. Assessing the degree of utility perceived by users and how they enjoy landscape services might help to reveal the multifunctional character of campus outdoor areas.

Land planners should consider the final beneficiary of the campus open space, which is the university community. Human beings are anthropocentric, so the design of the campus' environment should be adapted to their different profiles to satisfy their needs. A sustainable campus' outdoor environment should consider the short-term uses and the daily habits of university community members according to age, sex, relationship with UPV, and time spent on campus. Personal factors, such as interests, branch of knowledge, needs, and preferences, are important to understand their decision to use campus open spaces. Knowledge focus reveals a different degree of environmental concern about the quality of campus' open space in the provision of landscape services.

A proper distribution of campus open spaces provides equal availability and accessibility to the outdoors to the whole university community. However, physical connectivity is only the first step in assuring an effective landscape system: a reconnection with nature is needed, where ecological flows are facilitated and species networks created. The closest thing in a campus outdoor environment to a natural ecosystem is high biodiversity. More differently aged trees, shrubs, and bushes should be integrated into the campus landscape and connected with existing habitats to increase strata diversity. Shrubs and bodies of water are ecological stepping stones required by natural ecosystems and can be helpful in campus outdoor areas.

Our results related to the perceived high-quality benefits are only estimated through the survey study. Further quantitative analyses to provide a more objective assessment, compared to perception, are necessary for landscape planners to create a complex landscape, centred on the needs and preferences of the university community. A reliable system of green indicators to assess the quality with which

landscape services are provided by green outside areas should be considered for current design and future improvement actions.

Our research could help UPV meet the environmental guidelines inspired by the Green Flag's prerequisites. This study has ascertained the degree of satisfaction of the university community with the current conditions of campus open spaces to discover what the university community's daily needs are and to identify design criteria related to the provision of high-quality landscape services. In our opinion, the outdoor areas should inspire an excellent educational environment in the context of a healthy and sustainable place.

This study could be considered helpful in diagnosing possible improvements for the campus open spaces not only for the Universitat Politècnica de València, but also for other university campuses which are struggling to achieve sustainability and aspire to receive the Green Flag Award.

**Author Contributions:** Conceptualization, C.A.-M.T., M.V.-P., E.G., F.G. and R.A.; Data curation, C.A.-M.T. and R.A.; Investigation, C.A.-M.T.; Methodology, C.A.-M.T., M.V.-P., E.G., F.G. and R.A., Project administration, F.G. and M.V.-P.; Software, C.A.-M.T.; Supervision, M.V.-P., E.G. and F.G.; Writing—original draft, C.A.-M.T.; Writing—review & editing, M.V.-P., E.G., F.G. and R.A. All authors have read and agreed to the published version of the manuscript.

**Funding:** This research was funded by European Commission, European Union's Horizon 2020, Green Cities for Climate and Water Resilience, Sustainable Economic Growth, Healthy Citizens and Environments; grant number 730283.

**Acknowledgments:** We are grateful to the Universitat Politècnica de València for providing data and helping to distribute the questionnaire to the university community. We would also like to thank all the respondents for participating in the survey. This work was supported by the European Union's Horizon 2020 research and innovation programme under the project Green Cities for Climate and Water Resilience, Sustainable Economic Growth, Healthy Citizens and Environments, with the reference 730283.

**Conflicts of Interest:** The authors declare no conflict of interest.

## Appendix A

**Table A1.** Percentage of campus open spaces needs perceived by the university community.

| Needs | Sex | | Occupation | | | Knowledge | | Age | | | | Time Spent in UPV | | | | Frequency of COS | | |
|---|---|---|---|---|---|---|---|---|---|---|---|---|---|---|---|---|---|---|
| | Woman | Man | Student | AdSS | TRS | LRD | OD | 18–22 | 22–30 | 30–50 | 50 | ≤1 | 2–5 | 6–10 | >10 | Low | Medium | High |
| Furniture | 66.1 | 48.9 | 63.3 | 59.1 | 38.1 | 55.8 | 58.9 | 59.6 | 68.2 | 59.9 | 43.5 | 54.8 | 62.4 | 73.5 | 51.7 | 61.6 | 52.3 | 57.7 |
| Nearby COS | 32.1 | 29.1 | 34.2 | 25.8 | 24.5 | 30.2 | 31.3 | 41.4 | 27.0 | 25.8 | 21.8 | 35.5 | 36.6 | 31.4 | 26.0 | 25.0 | 37.5 | 31.4 |
| Water | 63.2 | 58.2 | 65.3 | 54.1 | 55.4 | 60.7 | 61.9 | 71.7 | 60.1 | 51.6 | 57.8 | 62.1 | 68.8 | 64.7 | 56.0 | 60.5 | 60.9 | 61.7 |
| Recreational | 18.0 | 19.0 | 20.0 | 17.0 | 13.7 | 15.9 | 20.2 | 22.5 | 16.2 | 20.3 | 12.2 | 18.5 | 23.7 | 13.7 | 16.6 | 18.8 | 15.6 | 19.5 |
| Tranquillity | 18.3 | 31.5 | 25.7 | 22.0 | 26.6 | 26.6 | 24.7 | 26.1 | 23.0 | 26.9 | 25.2 | 21.0 | 26.9 | 27.5 | 25.7 | 28.5 | 18.8 | 25.9 |
| Security | 5.9 | 6.5 | 4.8 | 8.8 | 7.2 | 7.5 | 5.3 | 5.0 | 2.7 | 9.9 | 6.1 | 3.2 | 6.5 | 2.9 | 7.4 | 5.2 | 7.8 | 6.1 |
| Classrooms | 49.9 | 36.1 | 50.2 | 37.1 | 26.6 | 51.6 | 38.5 | 50.0 | 51.4 | 39.0 | 31.3 | 46.0 | 43.5 | 65.7 | 36.0 | 38.5 | 50.8 | 43.3 |
| Less asphalt | 40.6 | 43.5 | 39.9 | 49.1 | 44.6 | 44.5 | 41.5 | 33.2 | 47.3 | 53.3 | 40.8 | 28.2 | 34.4 | 53.9 | 48.0 | 41.3 | 38.3 | 44.4 |
| Cleaner air | 27.5 | 34.8 | 33.4 | 27.7 | 28.1 | 33.1 | 30.2 | 31.8 | 37.2 | 28.0 | 29.3 | 28.2 | 29.0 | 42.2 | 31.1 | 29.7 | 28.1 | 32.8 |
| Trees | 53.5 | 61.7 | 53.9 | 62.9 | 65.5 | 60.1 | 56.4 | 49.6 | 59.5 | 64.8 | 61.9 | 47.6 | 48.4 | 65.7 | 63.7 | 62.8 | 58.6 | 55.9 |
| Natural shade | 68.9 | 68.8 | 68.1 | 71.9 | 75.8 | 71.1 | 67.7 | 67.1 | 70.9 | 67.6 | 73.5 | 66.9 | 67.2 | 66.7 | 70.9 | 75.0 | 66.4 | 67.6 |
| Native species | 32.6 | 35.1 | 35.1 | 35.2 | 32.4 | 39.3 | 31.9 | 34.6 | 35.1 | 34.6 | 34.0 | 27.4 | 31.2 | 51.0 | 34.3 | 34.3 | 36.7 | 34.3 |

## Appendix B

**Table A2.** Percentage of perceived landscape services provided by campus open spaces considering the current state and management of outdoor areas.

| Landscape Services | Sex | | Occupation | | | Knowledge | | Age | | | | | Time Spent in UPV | | | COS Frequency | | |
|---|---|---|---|---|---|---|---|---|---|---|---|---|---|---|---|---|---|---|
| | Woman | Man | Student | AdSS | TRS | LRD | OD | 18–20 | 22–30 | 30–50 | >50 | ≤1 | 2–5 | 6–10 | >10 | Low | Medium | High |
| Walks | 82.8 | 79.2 | 80.5 | 80.4 | 82.5 | 76.0 | 83.7 | 84.8 | 76.7 | 79.1 | 78.9 | 84.4 | 81.0 | 76.5 | 80.8 | 78.6 | 74.0 | 83.7 |
| Study/work | 54.5 | 43.6 | 51.5 | 42.2 | 48.8 | 48.8 | 49.2 | 56.6 | 43.8 | 49.7 | 38.0 | 65.0 | 51.6 | 47.5 | 42.9 | 43.3 | 44.1 | 52.6 |
| Daily needs | 66.8 | 50.7 | 61.1 | 58.7 | 50.4 | 56.3 | 60.4 | 65.0 | 58.2 | 59.3 | 46.3 | 67.8 | 58.8 | 66.3 | 54.0 | 52.5 | 54.3 | 63.0 |
| Relax | 84.1 | 79.0 | 83.2 | 78.7 | 80.2 | 77.9 | 83.9 | 84.0 | 82.9 | 81.0 | 76.2 | 86.0 | 80.4 | 79.4 | 81.2 | 81.1 | 80.3 | 82.6 |
| Play | 56.9 | 47.6 | 55.8 | 50.0 | 42.5 | 48.8 | 53.9 | 59.9 | 48.3 | 49.1 | 42.2 | 64.7 | 58.7 | 53.9 | 42.9 | 46.8 | 50.4 | 54.5 |
| Sport | 73.1 | 66.3 | 71.5 | 67.1 | 66.1 | 66.6 | 71.6 | 73.8 | 69.9 | 68.0 | 63.6 | 79.2 | 69.4 | 72.5 | 66.1 | 67.9 | 73.8 | 69.2 |
| Research | 31.2 | 19.0 | 28.9 | 21.3 | 14.7 | 27.0 | 23.4 | 31.7 | 26.7 | 21.2 | 14.3 | 47.9 | 26.0 | 23.5 | 17.8 | 12.4 | 28.2 | 28.5 |
| Learn | 44.3 | 31.0 | 38.8 | 42.2 | 29.4 | 42.2 | 34.8 | 41.0 | 35.2 | 34.7 | 37.7 | 55.0 | 38.0 | 31.7 | 33.8 | 28.8 | 40.3 | 40.3 |
| Art | 61.3 | 49.6 | 59.5 | 57.8 | 38.8 | 55.1 | 56.2 | 63.7 | 57.2 | 48.6 | 46.9 | 68.3 | 61.3 | 57.4 | 48.4 | 41.7 | 61.3 | 59.3 |
| Meeting | 90.5 | 78.8 | 87.0 | 84.2 | 78.0 | 83.4 | 85.5 | 89.8 | 84.2 | 83.7 | 76.5 | 90.9 | 87.4 | 85.3 | 81.1 | 82.6 | 81.0 | 86.6 |
| Floods | 19.7 | 21.1 | 22.2 | 14.2 | 17.7 | 19.5 | 20.4 | 21.0 | 22.9 | 21.7 | 12.9 | 28.0 | 19.8 | 20.0 | 16.2 | 14.9 | 23.0 | 21.2 |
| Air | 75.1 | 69.4 | 74.3 | 74.5 | 62.3 | 68.2 | 74.1 | 76.9 | 74.7 | 68.9 | 65.5 | 79.2 | 73.2 | 74.0 | 68.9 | 69.1 | 70.2 | 74.0 |
| Temp./light | 78.3 | 73.8 | 77.2 | 76.6 | 71.4 | 72.5 | 78.0 | 81.8 | 72.6 | 72.9 | 73.2 | 83.5 | 76.5 | 76.5 | 73.6 | 71.9 | 73.0 | 78.2 |
| Species | 37.4 | 28.4 | 31.4 | 41.9 | 28.8 | 33.8 | 33.0 | 33.0 | 28.0 | 33.5 | 39.8 | 36.7 | 32.4 | 31.7 | 31.6 | 29.0 | 27.6 | 36.6 |

## Appendix C

**Table A3.** Percentage of perceived landscape services provided by park.

| Landscape Services | Sex | | Occupation | | | Knowledge | | Age | | | | Time spent in UPV | | | | Frequency of COS | | |
|---|---|---|---|---|---|---|---|---|---|---|---|---|---|---|---|---|---|---|
| | Woman | Man | Student | AdSS | TRS | LRD | OD | 18–22 | 22–30 | 30–50 | >50 | ≤1 | 2–5 | 6–10 | >10 | Low | Medium | High |
| Pass by | 48.3 | 52.9 | 49.7 | 50.3 | 51.1 | 50.0 | 49.6 | 56.1 | 43.9 | 42.3 | 51.0 | 51.6 | 51.6 | 52.9 | 47.4 | 48.8 | 57.0 | 48.7 |
| Work/Study | 66.3 | 51.5 | 67.2 | 50.3 | 37.4 | 63.0 | 55.5 | 73.2 | 58.1 | 53.8 | 38.8 | 73.4 | 71.0 | 68.6 | 44.3 | 52.9 | 60.2 | 60.0 |
| Eat | 86.1 | 74.9 | 85.0 | 80.5 | 63.3 | 82.1 | 78.9 | 83.2 | 89.9 | 84.6 | 60.5 | 84.7 | 83.3 | 88.2 | 74.3 | 77.3 | 79.7 | 81.8 |
| Relax | 94.3 | 92.9 | 94.4 | 93.7 | 88.5 | 92.5 | 93.6 | 93.6 | 96.6 | 92.9 | 90.5 | 87.9 | 95.7 | 97.1 | 92.6 | 90.7 | 92.2 | 94.6 |
| Play | 74.6 | 59.1 | 70.6 | 67.3 | 54.0 | 69.8 | 65.3 | 70.7 | 71.6 | 73.1 | 51.0 | 70.2 | 71.0 | 68.6 | 64.0 | 62.8 | 68.0 | 68.8 |
| Walk/Do sport | 64.8 | 58.9 | 60.1 | 66.0 | 62.6 | 62.0 | 61.3 | 62.9 | 56.8 | 59.9 | 65.3 | 66.1 | 60.2 | 55.9 | 62.6 | 57.0 | 61.7 | 63.6 |
| Education | 60.4 | 51.5 | 58.7 | 59.1 | 42.4 | 57.8 | 54.5 | 58.2 | 56.8 | 59.9 | 45.6 | 60.5 | 58.6 | 58.8 | 53.1 | 52.3 | 57.0 | 57.1 |
| Research | 34.7 | 22.9 | 33.8 | 26.4 | 12.9 | 28.9 | 28.3 | 36.8 | 31.8 | 24.2 | 17.7 | 37.1 | 38.2 | 37.3 | 18.6 | 25.0 | 22.7 | 31.8 |
| Inspiration | 65.8 | 53.7 | 64.3 | 62.3 | 39.6 | 59.4 | 59.6 | 62.9 | 70.3 | 58.2 | 44.9 | 64.5 | 63.4 | 69.6 | 52.6 | 57.0 | 60.9 | 60.0 |
| Psychological connections | 55.8 | 45.5 | 52.8 | 54.1 | 39.6 | 51.6 | 50.0 | 52.5 | 51.4 | 51.6 | 43.5 | 53.2 | 51.6 | 57.8 | 46.9 | 47.1 | 50.0 | 52.7 |
| Meeting | 72.2 | 59.9 | 68.3 | 69.2 | 54.7 | 66.2 | 66.0 | 71.8 | 67.6 | 64.8 | 57.8 | 69.4 | 68.3 | 70.6 | 62.3 | 65.1 | 68.8 | 65.9 |
| Flood control | 22.4 | 20.7 | 22.3 | 21.4 | 18.0 | 21.4 | 21.5 | 17.1 | 31.8 | 23.6 | 16.3 | 12.9 | 20.4 | 30.4 | 21.1 | 19.8 | 25.8 | 20.5 |
| Air quality | 82.5 | 74.9 | 78.3 | 87.4 | 69.8 | 81.2 | 77.0 | 78.2 | 79.7 | 79.1 | 78.2 | 77.4 | 79.0 | 78.4 | 79.7 | 75.6 | 82.0 | 78.5 |
| Climatic conditions | 69.7 | 58.9 | 63.9 | 69.8 | 59.0 | 64.9 | 63.8 | 62.9 | 68.9 | 66.5 | 59.9 | 60.5 | 62.9 | 70.6 | 64.0 | 62.2 | 63.3 | 65.1 |
| Biodiversity | 67.1 | 56.4 | 57.4 | 67.3 | 50.4 | 58.1 | 57.9 | 59.6 | 54.7 | 61.5 | 53.1 | 57.3 | 62.4 | 54.9 | 56.3 | 52.9 | 56.3 | 60.3 |

## Appendix D

**Table A4.** Percentage of perceived landscape services provided by green verge.

| Landscape Services | Sex | | Occupation | | | Knowledge | | Age | | | | Time spent in UPV | | | | Frequency of COS | | |
|---|---|---|---|---|---|---|---|---|---|---|---|---|---|---|---|---|---|---|
| | Woman | Man | Student | AdSS | TRS | LRD | OD | 18–22 | 22–30 | 30–50 | >50 | ≤1 | 2–5 | 6–10 | >10 | Low | Medium | High |
| Pass by | 87.4 | 88.9 | 85.0 | 91.8 | 94.2 | 89.6 | 86.4 | 83.9 | 87.2 | 92.3 | 90.5 | 81.5 | 83.9 | 86.1 | 91.7 | 89.5 | 92.2 | 86.0 |
| Work/Study | 11.3 | 8.7 | 11.5 | 8.8 | 5.0 | 11.4 | 8.9 | 10.4 | 10.8 | 11.5 | 6.8 | 18.5 | 8.1 | 10.8 | 7.1 | 8.7 | 8.6 | 10.5 |
| Eat | 32.9 | 25.0 | 31.9 | 25.2 | 18.7 | 28.9 | 28.1 | 32.9 | 29.7 | 28 | 21.1 | 34.7 | 30.1 | 37.3 | 21.4 | 27.9 | 24.2 | 29.3 |
| Relax | 33.9 | 29.6 | 33.8 | 27.0 | 26.6 | 27.9 | 33.4 | 37.1 | 25.0 | 33.0 | 26.5 | 35.5 | 34.4 | 30.4 | 27.7 | 29.1 | 29.7 | 32.2 |
| Play | 8.7 | 7.3 | 8.8 | 6.9 | 5.0 | 9.1 | 7.2 | 9.6 | 8.1 | 8.8 | 3.4 | 10.5 | 8.6 | 9.8 | 6.3 | 8.7 | 10.2 | 7.1 |
| Walk/Do sport | 44.5 | 36.7 | 38.8 | 43.4 | 40.3 | 41.6 | 38.9 | 38.9 | 39.9 | 39.6 | 45.5 | 41.9 | 38.7 | 36.3 | 42.0 | 35.5 | 43.0 | 41.0 |
| Education | 14.4 | 9.0 | 11.7 | 12.6 | 10.1 | 14.3 | 9.8 | 10.7 | 13.5 | 13.7 | 8.2 | 15.3 | 10.8 | 11.8 | 11.1 | 14.5 | 12.5 | 10.3 |
| Research | 8.2 | 3.8 | 6.5 | 6.3 | 3.6 | 5.8 | 6.0 | 6.8 | 5.4 | 6.6 | 4.8 | 9.7 | 5.9 | 5.9 | 4.9 | 7.0 | 6.3 | 5.4 |
| Inspiration | 22.1 | 17.1 | 21.3 | 20.8 | 11.5 | 18.2 | 20.2 | 22.5 | 18.9 | 19.2 | 15.0 | 24.2 | 22.6 | 21.6 | 14.9 | 15.7 | 17.2 | 21.1 |
| Psychological connections | 18.3 | 13.6 | 16.3 | 15.1 | 13.7 | 13.0 | 17.4 | 15.0 | 16.9 | 15.9 | 15.6 | 17.7 | 15.1 | 19.6 | 14.6 | 14.0 | 18.8 | 15.7 |
| Meeting | 51.9 | 42.4 | 52.6 | 31.4 | 41.7 | 42.9 | 48.9 | 53.9 | 50.0 | 42.3 | 39.5 | 58.1 | 49.5 | 54.9 | 38.9 | 45.9 | 47.7 | 47.1 |
| Flood control | 4.6 | 6.8 | 6.3 | 3.8 | 5.0 | 6.2 | 5.1 | 7.9 | 5.4 | 3.8 | 3.4 | 7.3 | 6.5 | 4.9 | 4.3 | 5.2 | 7.0 | 5.2 |
| Air quality | 21.3 | 13.9 | 15.0 | 23.3 | 18.0 | 15.3 | 19.1 | 14.6 | 15.5 | 20.3 | 20.4 | 17.7 | 14.5 | 15.7 | 19.1 | 15.7 | 19.5 | 17.8 |
| Climatic conditions | 13.9 | 11.1 | 11.9 | 11.9 | 14.4 | 13.3 | 12.1 | 14.3 | 8.8 | 12.6 | 12.9 | 14.5 | 13.4 | 10.8 | 11.7 | 15.7 | 14.1 | 11.1 |
| Biodiversity | 14.1 | 7.9 | 11.3 | 12.6 | 5.8 | 10.4 | 11.1 | 12.5 | 10.1 | 9.9 | 8.2 | 11.3 | 14.0 | 10.8 | 8.6 | 10.5 | 11.7 | 10.7 |

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
