# Peer review of "Towards a Greener University: Perceptions of Landscape Services in Campus Open Space"

_sustainability, doi:10.3390/su12156047_

Round 1

Reviewer 1 Report

This is an extremely well structured, well argued evidence based paper that provides interesting and significant findings.  

Below I raise a few minor issues that could be addressed. 

Firstly, use of the terms 'real urban ecosystems' on line 58 seems out of place given the narrative of the paper.  What 'real' urban ecosystems are is not clear.

Secondly, including and architecture and art as 'environment' disciplines seems a bit questionable and should perhaps be explained more.

Finally, the last line of the conclusion seems somewhat out of place given the theme of the paper.  Perhaps remove?

Author Response

Response to Reviewer 1 Comments

Thank you for giving us the opportunity to submit a revised draft of the manuscript “Towards a Greener University: Perceptions of Landscape Services in Campus Open” for publication in the Journal Sustainability. We appreciate the time and effort that you dedicated to providing feedback on our manuscript and we are grateful for the insightful comments on and valuable improvements to our paper. We have incorporated your suggestions. Those changes are highlighted within the manuscript. Please see below, in red, for a point-by-point response to the reviewers’ comments and concerns. All page numbers refer to the revised manuscript file with tracked changes.

Point 1: Firstly, use of the terms 'real urban ecosystems' on line 58 seems out of place given the narrative of the paper.  What 'real' urban ecosystems are is not clear.

Response 1: Thank you for pointing this out. The term "urban ecosystem" seems much more appropriate for this purpose. We have already changed it (line 50).

Point 2: Secondly, including and architecture and art as 'environment' disciplines seems a bit questionable and should perhaps be explained more.

Response 2: Thank you for this contribution. We agree with your suggestion. We have already done the changes in the manuscript. We have added some arguments to explain how university community’s groups were classified according to their knowledge branch focus. Also, we have substituted term "Environment" for "Landscape related disciplines" throughout the paper. In some cases (tables: 6, 11, 13; annexes: 1,2,3,4) appears like "LRD" because of the extension of columns. For the same reason, class "Other studies" was replaced by "Other disciplines" and in tables appears like "Other" or "OD".

Proposal (lines:267-272): Finally, 308 persons (39.1%) come from landscape related disciplines (LRD) (e.g. forestry, agriculture, environmental engineering, architecture and fine arts), and 470 persons (59.7%) belong to other disciplines (OD) (e.g. economics, computer science, civil engineering, etc. The connection of the first group of disciplines with landscape through its analysis, planning, intervention or representation is assumed to involve a higher sensitivity to environment and open space.

Point 3: Finally, the last line of the conclusion seems somewhat out of place given the theme of the paper.  Perhaps remove?

Response 3: We think this is an excellent suggestion. We have removed it.

Reviewer 2 Report

There is a difference between ecosystem services and landscape services. Ecosystem services include not only recognition of plants but also animals (small ones) that contribute to the biodiversity. Besides that ecosystem services have important aspects of understanding space concerning mycroclimate, suds (sustainable urban drainage systems)... 

If you are using only one aspect of ecosystem services that should be notes within the article!

Author Response

Response to Reviewer 2 Comments

Thank you for giving us the opportunity to submit a revised draft of the manuscript “Towards a Greener University: Perceptions of Landscape Services in Campus Open” for publication in the Journal Sustainability. We appreciate the time and effort that you dedicated to providing feedback on our manuscript and we are grateful for the insightful comments on and valuable improvements to our paper. We have incorporated your suggestions. Those changes are highlighted within the manuscript. Please see below, in red, for a point-by-point response to the reviewers’ comments and concerns. All page numbers refer to the revised manuscript file with tracked changes.

Point 1: There is a difference between ecosystem services and landscape services. Ecosystem services include not only recognition of plants but also animals (small ones) that contribute to the biodiversity. Besides that, ecosystem services have important aspects of understanding space concerning microclimate, suds (sustainable urban drainage systems).

If you are using only one aspect of ecosystem services that should be notes within the article!

Response 1: Thank you for pointing this out. The term “biodiversity” includes more than vegetation, it encompasses all kind of species and also habitats. The aims and scale of the research allows us analyse visually the preferences and perceptions of university community of campus open space, from the point of view of superior vegetation species. We decide to use the apparent vegetation structure which is immovable and easily noticeable, rather than (small) animals, which are dynamic species. In our campus, bird species have started to awake the interest of campus environment unit. Future studies will include small animals for a more complex analysis of campus biodiversity. We have already done the changes in the text, by mentioning in a phrase what it is understood by “biodiversity” in our paper.  This way seems much more appropriate for this purpose. In table 2, we have added term "vegetation", so as not create any confusions among readers.

We agree that there are differences between ecosystem and landscape services.  That is why we decide to use landscape services concept instead of ecosystem services concept in the university campus arena (see lines: 60-69). Near vital and social benefits, we have included another example of landscape service -"habitats for biodiversity"

In table 1, theme regulation and maintenance of landscape services covers biotic (diversity of species and habitats) and physical environment. As reviewer mentions, ecosystem services tackle regulation of microclimate and suds (sustainable urban drainage systems). The item used in section 2, lines 206-207 ("It is useful for the management of environmental functions (biodiversity, permeability, climate comfort, CO2 capture") mentions biotic and physical functions of urban campus open space. Also, the items of two questions of the survey were designed to express benefits for two campus open spaces (park and green verge), offering the possibility to participants to choose some perceived landscape services related to biodiversity, microclimate, air quality and flood control (annex 3 and annex 4).

Proposal (lines: 60-65):

In this study campus open spaces are considered multifunctional landscape services (LS) providers, which simultaneously offer vital benefits, such as improving air quality, ensuring pleasant climatic conditions, habitats for biodiversity, offering outdoor spaces to play, do sport, rest, and provide psychological benefits. Although "biodiversity "encompasses all kind of species (including small animals) and also habitats, the aims and the scale of this study makes us consider only the superior plant vegetation for their apparent and easily recognizable structure in the campus.

Reviewer 3 Report

Dear authors, 

Congratulations on your work. This study is a result of a large quantitative inquiry that took place on your University Campus and should be important for designers and administration trying to improve the space in the future.
I can't question the surveys value as a study on this particular campus and it's users but I'm quite unsure about the value of the study in a larger context. 
The study is designed in a scientifically sound way but at the same time it's designed to give quite predictable results. This predictability is best characterised by two short sentences in "Results": "The most preferred COS are garden and park" and "On the opposite end of the spectrum, green verge and atrium are the least preferred". Based on 4 photographs your research has proved that general public prefers "green" landscapes over "grey" (mostly concrete, paved) landscapes. 

The article might be improved in several ways. First of all I'd like to see improved Introduction section.
The first paragraph (lines 33-41) is so general, broad, generic it is rather unnecessary. I would recommend removing it.
The introduction should also refer to previously published articles on visual quality/preference assessments in university campus environments. 
example:
Polat, Zöhre & Kilicaslan, Cigdem & Kara, Baris & Deniz, Bülent. (2015). Visual quality assessment of trees and shrubs in the south campus of Adnan Menderes university in spring. Fresenius Environmental Bulletin. 24. 4303-4315.

I would like you to explain in the Methodology section, why you only used 4 photographs of UGI elements (Figure 3). Visual Quality Assessments Methods usually use much larger amount of photographs to better present the complexity of the landscape. 

The study lacks some nuances. Example: you describe in the discussion section (4.2. Greener or greyer COS community preferences) that users of the campus "ask for more ‘green-blue’ elements, such as more natural shade, more trees, and more water". This is another predictable/general statement. How much more? Would it beneficial to the campus to have 100% park areas outside of the buildings? What is the correct ratio here? I would expect an inquiry like this to try to assess the expected/preferred ratio between green space and "urban" hard space, rather than stating that people prefer green over grey.

This nuancing could have been achieved by adding qualitative methods to the research design. It would be highly beneficial of the survey was accompanied by several in-depth interviews or focus groups. The lack of supporting qualitative methods is something I would consider as a "Limitation of the study" and add to that section accordingly.

Author Response

Response to Reviewer 3 Comments

Thank you for giving us the opportunity to submit a revised draft of the manuscript “Towards a Greener University: Perceptions of Landscape Services in Campus Open” for publication in the Journal Sustainability. We appreciate the time and effort that you dedicated to providing feedback on our manuscript and we are grateful for the insightful comments on and valuable improvements to our paper. We have incorporated your suggestions. Those changes are highlighted within the manuscript. Please see below, in red, for a point-by-point response to the reviewers’ comments and concerns. All page numbers refer to the revised manuscript file with tracked changes.

Point 1: First of all, I'd like to see improved Introduction section.

The first paragraph (lines 33-41) is so general, broad, generic it is rather unnecessary. I would recommend removing it.

Response 1: Thank you for this contribution. The reviewer is correct. Our intention was to make the transition between the most common green space, which have ever gained Green Flag Award, and the recently incorporation of green campuses. We have removed the first paragraph (lines 33-41), as we focus on green campuses.

Point 2: The introduction should also refer to previously published articles on visual quality/preference assessments in university campus environments.

example: Polat, Zöhre & Kilicaslan, Cigdem & Kara, Baris & Deniz, Bülent. (2015). Visual quality assessment of trees and shrubs in the south campus of Adnan Menderes university in spring. Fresenius Environmental Bulletin. 24. 4303-4315.

Response 2: Thank you for your suggestion and for sharing us a related article, whose reference we have already included in our research. The investigation of Polat et al. 2015 is very helpful and interesting. Maybe we will apply this method for further studies in our campus.

Proposal (lines 87-89): Concerning visual quality assessment (VQA) in university campus environments, Polat et al. [16] demonstrate that visual quality increases in the areas where naturalness and plants are dominant, especially trees and shrubs, which increase also students ‘preferences for landscapes.

Point 3: I would like you to explain in the Methodology section, why you only used 4 photographs of UGI elements (Figure 3). Visual Quality Assessments Methods usually use much larger amount of photographs to better present the complexity of the landscape.

Response 3: Thank you for pointing this out. More photographs are needed in order to appreciate correctly the perceptions of people and choose photographs to better present the complexity of landscapes. We agree that accuracy and reliability are always important elements in any research. As urban landscape/ecosystem is the scope of the study and includes quite homogeneous elements, we understand that those four photographs are the minimum sample to cover the variability of types of campus outdoor spaces from the point of view of the potential of providing landscape services. The objective of this research was not to qualify all campus open spaces, but to assess the perceptions of university community of the power of the current typology of campus open space of providing benefits and find differences between them. Moreover, campus is lacking of natural spaces, such as: forests, shrubland or orchards, which could have made the difference between the perceptions of other types of outdoor environment and the urban ones.

Also, we are conscious that the questionnaire is a bit large (more than 100 variables). In this research we used only a part of them. Thus, we tried to avoid to put university community in the situation of answering to more questions or make larger comparations.

We have already included in the article the explanation that reviewer required about using of only four photographs.

Proposal (lines: 175-183):

 Colour photographs are commonly used in studies of perceptions and judgment of the visual environment [13], [27], [30], [31], and for this study, four images of different UGI elements were included. Open spaces with different green cover and permeable areas were selected: atrium, garden, park, and green verge (Figure 3). These were selected to represent the different building stages of university campus. Atrium is the newest campus area, garden was included in the initial design of campus, and the other two elements were built in the intermediary stage. Also, they portray the most popular areas in the campus and meet the condition of including different elements of urban green infrastructure. Then, four photographs are the minimum sample to cover the variability of campus outdoor environment regarding provided landscape services. For the classification of UGI, EU criteria [32] was used. Their organisation by location and use was inspired by Cooper and Wischemann’s campus outdoor environment classification [3].

Point 4: The study lacks some nuances. Example: you describe in the discussion section (4.2. Greener or greyer COS community preferences) that users of the campus "ask for more ‘green-blue’ elements, such as more natural shade, more trees, and more water". This is another predictable/general statement. How much more? Would it beneficial to the campus to have 100% park areas outside of the buildings? What is the correct ratio here? I would expect an inquiry like this to try to assess the expected/preferred ratio between green space and "urban" hard space, rather than stating that people prefer green over grey.

This nuancing could have been achieved by adding qualitative methods to the research design. It would be highly beneficial of the survey was accompanied by several in-depth interviews or focus groups. The lack of supporting qualitative methods is something I would consider as a "Limitation of the study" and add to that section accordingly.

Response 4: Thank you for this contribution. We agree with the reviewer’s suggestion. Reviewer points out the simplicity of the fact that respondents stated only their preferences between green and grey spaces. The suggestion of reviewer helps us to plan new and future more complex assessment studies. Also, thank you for the advice regarding the involvement of more qualitative methods for the future studies. We have added this part as "Limitation of the study" as the reviewer suggests.

Proposal (lines: 589-592):

This work seeks to know the benefits perceived by users of open spaces, focusing on two of the most popular ones. Future research will address more specific aspects for improving open spaces, e.g. expected/preferred ratio between green space and urban space. Qualitative techniques, such as: in-depth interviews and focus groups could be added in order to supplement quantitative methods.

Round 2

Reviewer 3 Report

Dear authors, 
Thank you for the answers provided and corrections made to the manuscript. I'm pleased with the improvement and I'm recommending for the article to be published after final editing.